# Impact of Fixed Cost Increase on the Optimization of Two-Stage Sustainable Supply Chain Networks

**Ahmed Mostafa** [1] **, Kamal Moustafa** [1] **and Raafat Elshaer** [2,*]

[1]    Industrial Engineering Department, Faculty of Engineering, Zagazig University, Zagazig 44519, Egypt; amatya@eng.zu.edu.eg (A.M.); kamostafa@eng.zu.edu.eg (K.M.)

[2]    Industrial Engineering Department, Collage of Engineering, King Khalid University, Abha 61421, Saudi Arabia

*    Correspondence: relshaer@kku.edu.sa

**Abstract:** Supply chain networks have emerged as the backbone of economic activities in the modern world. Most of the literature on the optimization problem of transportation in supply chain networks (SCNs) considers fixed costs (FCs). Although most researchers have used fixed costs in supply chain design, none have studied their impact on optimal supply chain design. The design of the network can change when we take fixed costs into optimization. However, in the practical application of this problem, it is important to study the effect of FCs. This paper examines the impact of the fixed cost's increase as well as the variable cost on the optimization of a two-stage SCN. To do so, two mathematical models for mixed-integer nonlinear programming were developed. The first model optimizes fixed and variable costs in addition to the opening cost, whereas the second model only optimizes variable and opening costs. To evaluate the effect of considering the FC on the optimization supply chain problem, four groups of instances were randomly generated and solved using Lingo. The results of the two models are compared using the average percentage deviation. In addition, sensitivity analysis was performed to determine the impact of changes in opening and variable costs on the considered optimization problem. The computational results and the sensitivity analysis show that the first model with minimized FC outperforms the second model, which does not consider the FC in minimization and FC affects the optimization.

**Keywords:** supply chain; network design; two-stage; fixed cost; sensitivity analysis

## 1. Introduction

The supply chain network design (SCND) problem can be classified into three types: forward type of supply chain network, reverse type of supply chain network, and closed loop type of supply chain network. In this study, we deal with the forward type of SCND, which is when the material flows from the supplier to the plant and then reaches the customer. There are many stages in this model (multistage), which may include warehouses and distributors. The supply chain is a network of suppliers, manufacturers, and distribution centers that transforms raw materials into usable products through several stages and distributes them to retailers in one or more stages. In the traditional transportation problem, the objective is to minimize total transportation costs by minimizing shipping costs proportional to the volume of items moved. Nevertheless, in practice, a fixed cost is incurred anytime a transportation route is established between a factory and a retailer. A fixed-cost transportation problem (FCTP) occurs when both fixed and variable costs exist simultaneously. The variable cost varies linearly with the amount carried from supply to destination, and the fixed cost is incurred anytime a product is transferred between the supply and destination points. The preceding evidence demonstrates that problems involving fixed costs are more challenging to resolve than those involving variable costs.

Typically, the FCTP is formulated as a mixed-integer programming problem and is tackled using methods comparable to those described in the literature. Several reports

on the single-stage FCTP emphasize minimizing total transportation costs [1–3]. Molla-Alizadeh-Zavardehi et al. [4] extended the FCTP to a two-stage supply chain problem, which considers possible distribution centers with fixed capacity for each distribution center to be opened. The model optimizes overall costs by establishing the optimal number of distribution centers to meet customer demands. Panicker et al. [5] proposed a two-stage FCTP in which several plants serve products to a number of retailers via a set of unlimited-capacity distribution centers. However, the model only dealt with a single product and a single period. Similarly, Hong et al. [6] considered the FCTP in a two-stage supply chain with fixed costs for transportation routes.

In practice, it is more challenging to solve the FCTP than to solve a linear one. Numerous techniques have been proposed in the literature to solve the FCTP, such as the genetic algorithm [7,8], the simulated annealing algorithm [9], the artificial immune and genetic algorithm [4], simplex-based simulated annealing [10], and ant colony optimization for various optimization problems such as the traveling salesman problem [11]. Jawahar and Balaji [12] considered a two-stage supply chain distribution problem associated with a fixed cost. Stützle and Dorigo [13] applied ant colony optimization to solving complex cases of the FCTP. Panicker et al. [5] conducted a comparative analysis of an ant colony optimization and a genetic algorithm heuristic technique, revealing their effectiveness in solving a two-stage FCTP. Sanei et al. [14] proposed a Lagrangian relaxation heuristic for solving the problem of transporting products from sources to destinations using different transportation modes with variable and fixed costs. Lotfi and Tavakkoli-Moghaddam [1] proposed a genetic algorithm to solve FCTPs. Shen and Zhu [15] examined a two-stage FCTP under uncertainty and proposed to use the genetic algorithm and particle swarm optimization to solve the problem. Kowalski et al. [3] developed a simple algorithm for obtaining the global solution to a small-scale FCTP by decomposing the problem into a series of smaller subproblems. In the real world, the FCTP is a complicated problem, particularly in supply chain management and distribution systems. Sadeghi-Moghaddam et al. [16] presented variable and fixed costs as fuzzy numbers. Panicker and Sarin [17] and Wang et al. [18] formulated a multistage, multiperiod FCTP for multiproducts, which is solved using ant colony optimization. In our previous paper, Mostafa and Elshaer [19] solved the two-stage FCTP using three ant colony optimization algorithms by comparing the Lingo optimum solution. This research aims to answer the following question: do fixed costs have an impact on the optimal design of supply chain networks?

The structure of this paper is as follows: Section 2 presents a literature review. In Section 3, the problem formulation is presented. In Section 4, the results and computational study are illustrated. Sensitivity analysis is described in Section 5. Section 6 provides conclusions and suggestions for future work.

## 2. Literature Review

The design of a supply chain network is one of the most complex topics in supply chain management, including decisions at the operational, tactical, and strategic levels. This problem entails identifying the number, location, and capacity of facilities, as well as establishing distribution channels and flows of materials and products that will be manufactured and transported to suppliers at each consumption layer.

The authors addressed the FCTP in one, two, or multiple stages, which can be classified as follows according to the number of stages, with indications of the cost components and solution methodology in each case:

### 2.1. One Stage FCTP

Lotfi and Tavakkoli-Moghaddam [1] proposed a genetic algorithm using priority-based encoding (pb-GA) for linear and non-linear FCTPs. Adlakha et al. [2] developed a heuristic algorithm to identify the demand destinations and the supply points to ship to. Another aspect to consider is the more-for-less (MFL) phenomenon in FCTPs. The MFL phenomenon occurs when it is possible to ship more total goods for less (or equal)

total cost while shipping the same quantity or more from each origin and to each destination. LINGO 19 was used to obtain the optimal solution and the MFL solution for comparative purposes.

### 2.2. Two Stage FCTP

Molla-Alizadeh-Zavardehi et al. [4] considered two stages of the supply chain network: distribution centers (DCs) and customers. Customers with specific needs exist, as do prospective locations for warehouses. Each of the possible DCs can ship to any of the clients. Two different types of costs were considered: the opening cost, which is expected for opening a possible DC, and the shipping cost per unit from the DC to the clients. The proposed model picks several viable locations as distribution centers in order to meet the needs of all clients. Two algorithms, the genetic algorithm and the artificial immune algorithm, were created to address the given problem. The Taguchi experimental design approach was used to identify the best parameters with the fewest number of experiments. Different problem sizes were used, and the computational output of the algorithms was compared to one another for the purpose of performance evaluation of the suggested algorithms.

Hong et al. [6] focused on the problem of distribution allocation in a supply chain with two stages and fixed costs. His challenge was to identify a distribution network's manufacturing facilities, wholesalers, and retailers' supply chain arrangements. The issue was modeled using an integer-programming approach. The mathematical model includes fixed costs for facility opening and fixed costs for transportation routes, as well as unit transportation costs between entities. The model's goal was to reduce the overall expenses of supply chain management associated with assigning retailers to distribution centers and distribution centers to production facilities. For the purpose of solving the model, an ant colony optimization (ACO)-based heuristic was created. On a range of produced problem sizes, the heuristic was tested.

Panicker et al. [5] focused on an issue of distribution allocation in a supply chain with two stages and fixed costs. To handle the problem of a fixed transportation cost for a route, an algorithm based on ant colony optimization was suggested. A numerical analysis of examples of benchmark problems was performed. The proposed algorithm's outcomes were contrasted with those of the genetic algorithm-based heuristic. The design and management of supply chains were found to be the key concerns for managers of industrial and service organizations in today's fiercely competitive business environment. Allocating customers to a manufacturing company's various supply chain partners is a crucial choice that influences value addition, degree of customer service, and prices.

Sanei et al. [14] introduced the step fixed-charge solid transportation problem, in which products are transported using a combination of unit and step fixed-charges from sources to destinations. It offers a dual decomposition method that can handle larger cases and relies on Lagrangian relaxation. Lotfi and Tavakkoli-Moghaddam [1] proposed a priority-based genetic algorithm to solve both linear and nonlinear FCTPs that includes novel crossover and mutation operators. Shen and Zhu [15] examined the two-stage fixed-cost transportation problem in an unpredictable environment. Demands, supply, availability, fixed costs, and transported amounts were all regarded as uncertain factors because there are so many unknowns. The goal was to maximize overall profit in unpredictable circumstances. Based on the structure of the problem, the genetic algorithm and particle swarm optimization were suggested to solve the equivalent forms of the models.

Kowalski et al. [3] presented a simple method to find the overall solution to an FCTP on a small scale. By breaking the problem down into a number of smaller subproblems, the suggested approach solved FCTP. To solve the small-scale FCTP, a straightforward and quick branching approach was suggested. Until a perfect solution is found, the subproblems can be divided into even smaller subproblems. This approach offers a substitute for quickly using computer tools to solve small-scale problems. As a result, it can be used as a pedagogical tool in a classroom setting to achieve educational objectives. Sadeghi-

Moghaddam et al. [16] presented variable and fixed costs as fuzzy numbers. They used both priority-based representation and Prüfer numbers based on spanning trees to propose novel techniques in solution algorithms. Additionally, the Taguchi method was employed to ensure the accurate calibration of parameters and the proper operation of algorithms. Additionally, a number of instances of various sizes were produced to evaluate the effectiveness of the algorithms and available software in the context of real-world cases. Keshavarz et al. [20] considered a fixed-charge transportation problem with fuzzy shipping costs where the shipping costs of routes are fuzzy intervals with increasing linear membership functions, fixed costs, supplies, and demands are deterministic numbers.

### 2.3. Multistage FCTP

In addition to the variable cost, the majority of practical applications of a transportation network have a fixed cost. For a scenario involving multiple products, the problem is described as a multistage, multiperiod fixed-charge transportation problem (MPFCTP). Panicker and Sarin [17] modeled the problem with the use of an optimization modeling tool named "A Mathematical Programming Language," and the BONMIN solver provides the answer. Finding the best solution for a huge problem size that is seen in practice typically requires more precise algorithms and longer computation times. A heuristic based on ant colony optimization was suggested for these operational problems, where process speed is just as crucial as solution quality. Using datasets created at random, the solution produced by the suggested heuristic was contrasted with that of accurate approaches. For a scenario involving many products, a simple MPFCTP model was created. The computational analysis shows that even though the solution reached using exact methods is the best, the computational time required to solve a mathematical model is substantial. The ant colony optimization approach was suggested to solve the model since the exact methods become less effective as the problem size increases. Both approaches are the subject of a computational investigation, and the outcomes of the precise method and the ACO were compared. It is known that ACO provides a solution that is close to optimal in a lot less time than exact approaches.

A market area receives a variety of products from the supply chain over time. The model takes into account where manufacturers and retailers are located and makes the assumption that customer behavior is probabilistic and based on an attraction function that is affected by both the location and the quality of the retailers. In order to maximize supply chain profit in a competitive economy, Wang et al. [18] studied a model of a supply chain network with pricing competition. They built the supply chain with capacity constraints. A model of mixed integer nonlinear programming was used to formulate the issue. Simulated annealing search (SA) and particle swarm optimization (PSO) were the two heuristic techniques they suggested. The results based on solving designed datasets demonstrated that simulated annealing is more effective than particle swarm optimization in terms of both solution quality and CPU times. Kungwalsong et al. [21] proposed a two-stage stochastic programming model for a four-echelon global supply chain network design problem, considering possible disruptions at facilities. A modified simulated annealing (SA) algorithm was developed to determine the strategic decision at the first stage.

In our previous work to develop two-stage supply chain networks to solve FCTPs, Mostafa and Elshaer [19] proposed three ant colony-based algorithms, ACO1, ACO2, and ACO3. ACO2 and ACO3 are based on the development of two new pheromone trails and one heuristic trail. The proposed algorithms are tested on the produced problem instances, with the results compared to those achieved using Lingo 19.

The literature demonstrates that the researchers consider the FCs when developing supply chains without studying their impact on optimization, which is the motivation of this study. The main objective of this paper is twofold: first, we investigate the impact of fixed costs on the optimal design of two-stage FCTPs. Second, sensitivity analysis is employed to determine the effect of changes in variable costs and opening costs on optimization models without changing FCs.

### 3. Problem Formulation

Our two-stage SC problem can be described as a node-connected graph (plants to distributors, 1st stage, and distributors to retailers, 2nd stage) with edges (routes) connecting these nodes, as shown in Figure 1. As shown in the figure, there are a set of $m$ plants, $I = \{1, 2, \ldots, i, \ldots, m\}$, a set of $d$ distributors, $J = \{1, 2, \ldots, j, \ldots, d\}$, and a set of $r$ retailers, $K = \{1, 2, \ldots, k, \ldots, r\}$. In each route, both variable and fixed transportation costs are included. And a fixed opening cost for each distributor is also incurred. The manufacturing plant is where the products are produced. The distribution centers are warehouses of different capacities that store the products before they are delivered to retailers using vehicles. Depending on the capacity permitted, final products are produced in any one of the manufacturing facilities, and distribution centers then ship the finished goods to the retailers. Distribution centers are defined as facilities that maintain inventory. The problem contains information on potential locations for intermediate distribution centers. There is an initial cost for establishing the distribution center at each location. In addition, the shipping costs are known throughout. Retailer demands have deterministic values and are predetermined for the subsequent planning horizon. It is assumed that the final products have to be delivered through one of the distribution centers to the retailers. The manufacturing capacity of the plant and the distribution center capacity are designed to be of a reasonable size to absorb the total demand. The objective of the problem is to allocate retailers to the distribution center and distribution centers to the manufacturing plant, minimizing the total costs of supply chain operations.

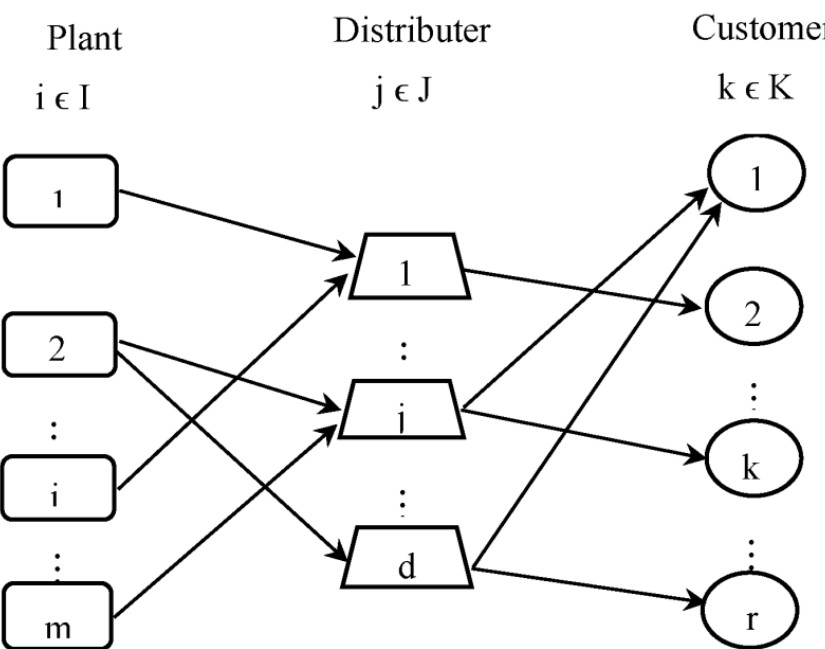

**Figure 1.** Graphical representation of the proposed network diagram [19].

As mentioned above, the main goal of this article is to investigate the impact of the FC increase on optimization. Therefore, the problem is formulated as mixed-integer nonlinear programming [19] by two models. The two models consider the minimization of two different objective functions to determine the optimum solution. The first model objective function is the sum of three types of costs: fixed, opening, and flow costs. The second model objective function, on the other hand, is the sum of the fixed cost and the minimum of the sum of the flow and the opening cost. The first model's objective, $Z_1$, is the minimization of three types of costs (fixed, opening, and flow), defined mathematically in Equation (1). While the second model's objective, $Z_2$, is the sum of the fixed cost and the minimized sum of the opening and flow costs, defined mathematically in Equation (2). The objective of the two models is to minimize the total cost involved in running a supply chain. It is clear that

the fixed cost is considered in the minimization in Equation (1), but in Equation (2) it is not considered.

The parameters, decision variables, models' objectives, and the models' constraints considered in the present work are defined and explained as follows:

Indices

$I$ Plants set, $i = 1, 2, \ldots, m$.
$J$ Distributors set, $j = 1, 2, \ldots, d$.
$K$ Retailers set, $k = 1, 2, \ldots, r$.

Parameters of capacity and demand

$S_i$ Plant $i$ capacity.
$SC_j$ Distributor $j$ Storage capacity.
$D_k$ Retailer $k$ demand.

Decision variables

$flPD_{ij}$ Transported units in 1st stage.
$flDC_{jk}$ Transported units in 2nd stage.
$xPD_{ij}$ Binary (specifies whether the units are transported in 1st stage).
$xDC_{jk}$ Binary (specifies whether the units are transported in 2nd stage).
$xD_j$ Binary (specifies whether a new distributor is open).

Cost parameters

$cPD_{ij}$ Unit transportation cost in 1st stage.
$cDC_{jk}$ Unit transportation cost in 2nd stage.
$fxPD_{ij}$ Fixed cost of transportation in 1st stage.
$fxDC_{jk}$ Fixed cost of transportation in 2nd stage.
$fxD_j$ Fixed opening cost for a new distributor.

$$Z_1 = Min \ (Fixed\ Cost + Flow\ Cost + Opening\ Cost) \tag{1}$$

$$Z_2 = Fixed\ Cost + Min \ (Flow\ Cost + Opening\ Cost) \tag{2}$$

The fixed, flow, and opening cost of Equations (1) and (2) are defined as follows:

$$Fixed\ Cost = \sum_{i=1}^{m} \sum_{j=1}^{d} (fxPD_{ij} \times xPD_{ij}) + \sum_{j=1}^{d} \sum_{k=1}^{r} (fxDC_{jk} \times xDC_{jk}) \tag{3}$$

$$Flow\ Cost = \sum_{i=1}^{m} \sum_{j=1}^{d} (cPD_{ij} \times flPD_{ij}) + \sum_{j=1}^{d} \sum_{k=1}^{r} (cDC_{jk} \times flDC_{jk}) \tag{4}$$

$$Opening\ Cost = \sum_{j=1}^{d} fxD_j \times xD_j \tag{5}$$

Subject to

$$\sum_{j=1}^{d} flPD_{ij} \ \leq S_i \qquad\qquad (i = 1, 2, \ldots, m) \tag{6}$$

$$\sum_{i=1}^{m} flPD_{ij} \ \leq \sum_{k=1}^{r} flDC_{jk} \ (j = 1, 2, \ldots, d) \tag{7}$$

$$\sum_{j=1}^{d} flDC_{jk} = D_k \qquad\qquad (k = 1, 2, \ldots, r) \tag{8}$$

$$\sum_{k=1}^{r} D_k = SC_j \qquad\qquad (j = 1, 2, \ldots, d) \tag{9}$$

$$\sum_{k=1}^{r} flDC_{jk} \ \leq \ SC_j \times xD_j \qquad (j = 1, 2, \ldots, d) \tag{10}$$

$$flPD_{ij}, \; flDC_{jk} \geq 0 \qquad \forall \, i, j, k \tag{11}$$

$$xPD_{ij} = \begin{cases} 1, & flPD_{ij} > 0 \\ 0, & otherwise \end{cases} \tag{12}$$

$$xDC_{jk} = \begin{cases} 1, & flDC_{jk} > 0 \\ 0, & otherwise \end{cases} \tag{13}$$

$$xD_j = \begin{cases} 1, & if \; distributor \; center \; j \; is \; open \\ 0, & otherwise \end{cases} \tag{14}$$

Constraints (6) denote that the quantity transported in the first stage is less than the plant capacity. Constraints (7) represent the balance of material in the two stages. Constraints (8) sets the quantity transferred to the retailer from the distributor equal to the retailer's demand. Constraints (9) limits the capacity of the distributor. The Constraints (10) set the quantity transported from the distributor to be less than or equal to its storage capacity. The Constraints (11) enforces that the decision variables $flPD_{ij}$ and $flDC_{jk}$ must be positive. The binary variables $xPD_{ij}$, $xDC_{jk}$ and $xD_j$ are presented in the Constraints (12)–(14).

## 4. Computational Study

In this study, the optimal solution is obtained by minimizing $Z_1$, and $Z_2$ and comparing them, subject to the same constraints as in [19]. The numerical instances are generated and simulated in Section 4.1. In Section 4.2, the computational results of solving the test instances using Lingo 19 are graphed and discussed.

### 4.1. Numerical Simulation

In order to investigate the effect of the FC increase on the proposed model, 1040 problems were generated randomly and classified into four groups (260 instances × 4 groups), as shown in Table 1. Each group had the same number of plants, distributors, and retailers. The generated demand of the retailers follows the uniform distribution. U (50, 500], and U (10,000, 20,000] were used for generating the distributors' opening costs. Table 2 illustrates the fixed and variable cost ranges in the testing problems of Mostafa and Elshaer [19]. In each instance of the generated problems, β routes were selected for changing their FC with δ = 0%, 10%,..., 50%. The number of routes, β, was randomly selected by 10%, 20%, 30%, 40%, and 50% from the total number of routes.

**Table 1.** Problem instance size.

| Problem Groups | Number of Instances | Number of Plants | Number of Distributors | Number of Retailers |
|---|---|---|---|---|
| G_1 | 260 | 2 | 5 | 10 |
| G_2 | 260 | 4 | 8 | 15 |
| G_3 | 260 | 6 | 10 | 20 |
| G_4 | 260 | 10 | 20 | 30 |

**Table 2.** Fixed and variable cost ranges.

| Fixed Cost Range | Variable Cost Range in 1st Stage | Variable Cost Range in 2nd Stage |
|---|---|---|
| (30–50) × Avg. variable cost | 10–30 | 10–50 |

### 4.2. Results and Discussion

The two models were solved using Lingo 19, with the first model $Z_1$ considering the FC in the optimization (see Appendix A) and the second model $Z_2$ not considering the

FC in the optimization (see Appendix B). The results were compared using the average percentage deviation in Equation (16), as demonstrated in Figures 2–5. According to the figures, the optimal solutions are improved when considering FC in the optimization, and the following is observed: (1) The average percentage of the total cost is enhanced by increasing the number of routes β whose FC increased. (2) For any specific change in the number of routes (x-axis) whose fixed costs will change, the percentage of improvement is directly proportional to the percentage change in FC. Therefore, the computational results shown in Figures 2–5 reveal that the optimal solutions of the first model $Z_1$ outperform the other optimal solutions of the second model $Z_2$, and the FC increase affects the optimality.

$$\% \; improvement = \frac{Z_2 - Z_1}{Z_2} \times 100\% \tag{15}$$

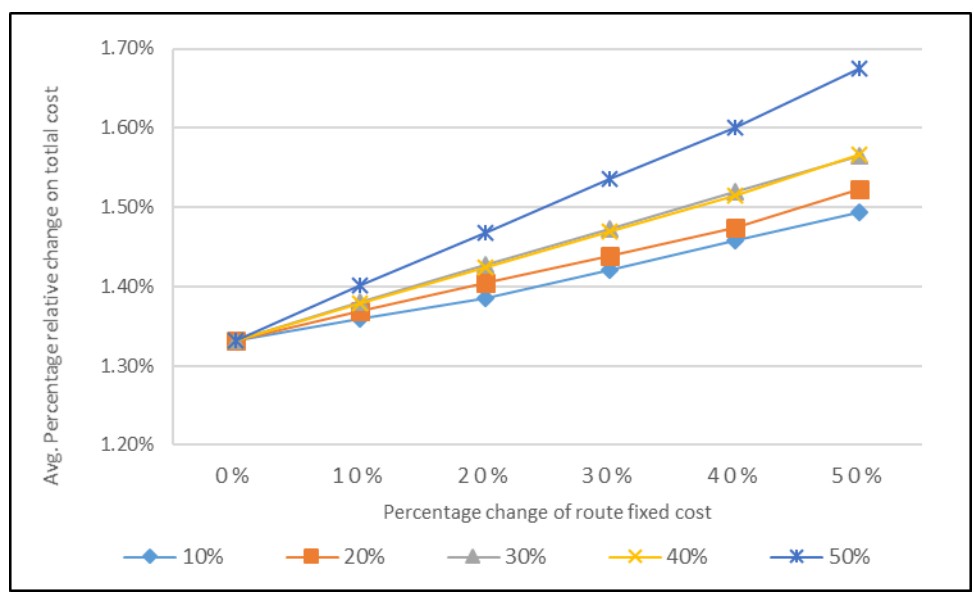

**Figure 2.** % improvement for the 1st group G_1 (2–5–10).

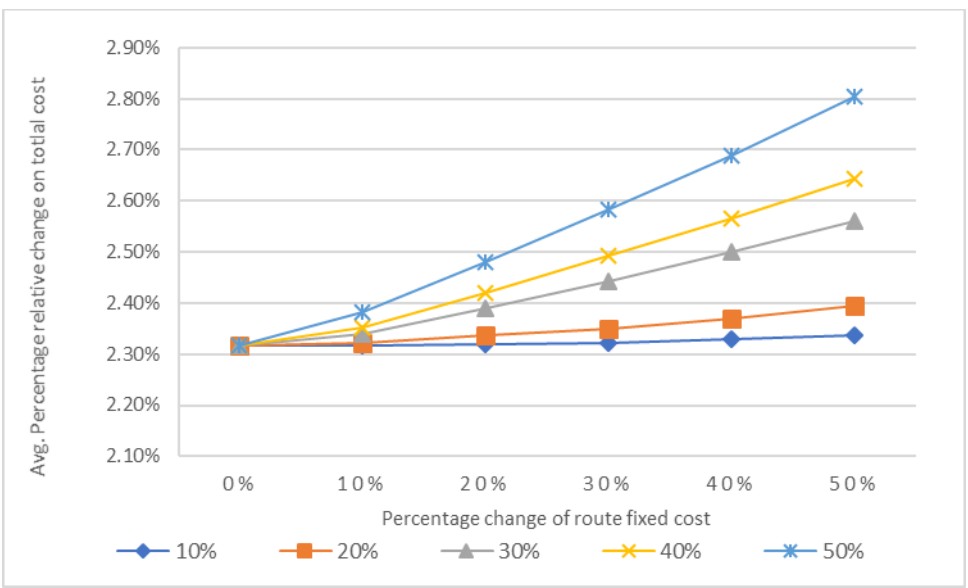

**Figure 3.** % improvement for the 2nd group G_2 (4–8–15).

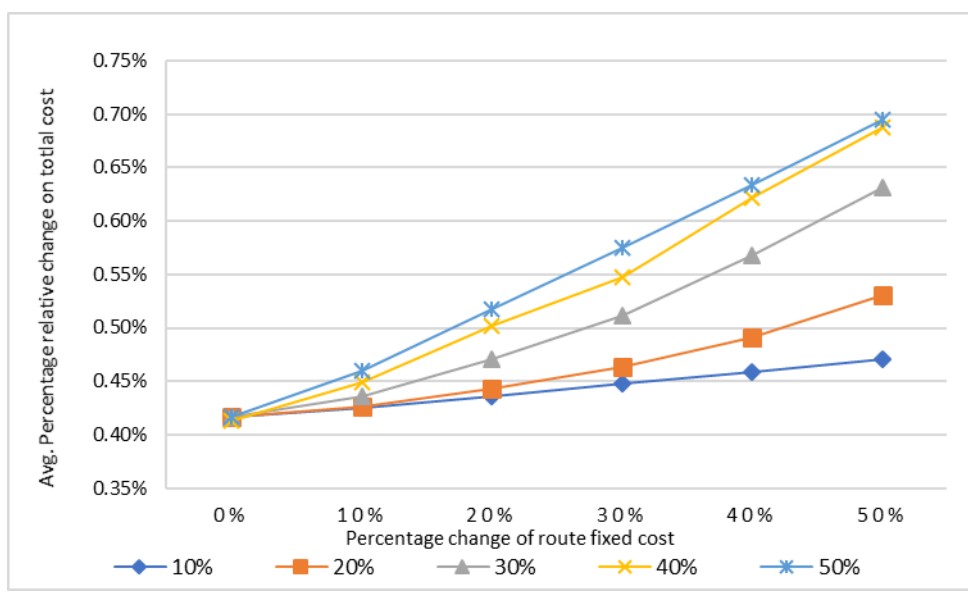

**Figure 4.** % improvement for the 3rd group G_3 (6–10–20).

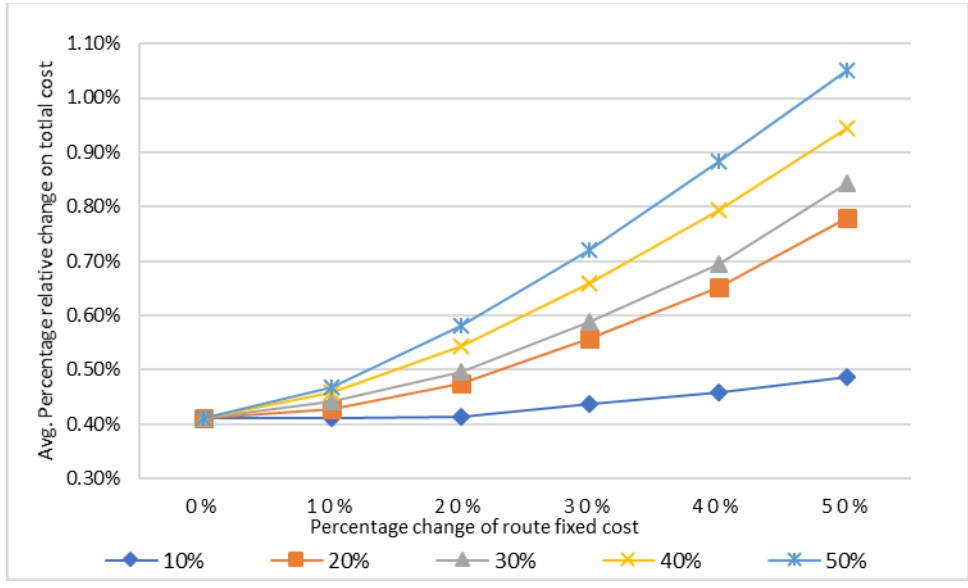

**Figure 5.** % improvement for the 4th group G_4 (10–20–30).

## 5. Sensitivity Analysis

The sensitivity analysis technique is used to investigate the impact of variable changes. This is the process by which it is determined how changes in variables (output) are caused by parameter changes. A significant result of sensitivity analysis refers to the factors that have the most significant impact on the design. In this paper, we investigate the impact of changes in opening and variable costs on the optimization of a two-stage supply chain network design problem. Therefore, two experiments were designed, as shown in the following two subsections.

### 5.1. Opening Costs

In order to investigate the impact of opening cost change on the two optimization models, problem instances with a particular size (6 plants, 10 distributors, 15 retailers) were generated uniformly from the ranges shown in Table 3. In the table, the opening cost was generated from the range [10,000 $(1 + \alpha)$, 20,000 $(1 + \alpha)$]. Where $\alpha$ takes the values from $-40\%$ to $40\%$, with $10\%$ steps. Thirty problem instances were generated for each $\alpha$

and solved optimally using the two models ($Z_1$ and $Z_2$) by Lingo. The average results of opening, fixed, flow, and total costs for the instances are depicted in Figure 6. The figure shows that model $Z_2$ which disregards FC in minimization, performed better on flow costs alone. However, model $Z_1$ outperformed model $Z_2$ in the opening, fixed, and total costs.

**Table 3.** Opening cost sensitivity analysis.

| Parameter | Range | |
| --- | --- | --- |
| Retailer demand | 50 | 500 |
| Opening cost of the distribution center. | $10,000\,(1+\alpha)$ | $20,000\,(1+\alpha)$ |
| Fixed cost = 30–50 × average variable cost | 30 | 50 |
| Variable cost = 10–30 in stage 1 | 10 | 30 |
| Variable cost = 10–50 in stage 2 | 10 | 50 |

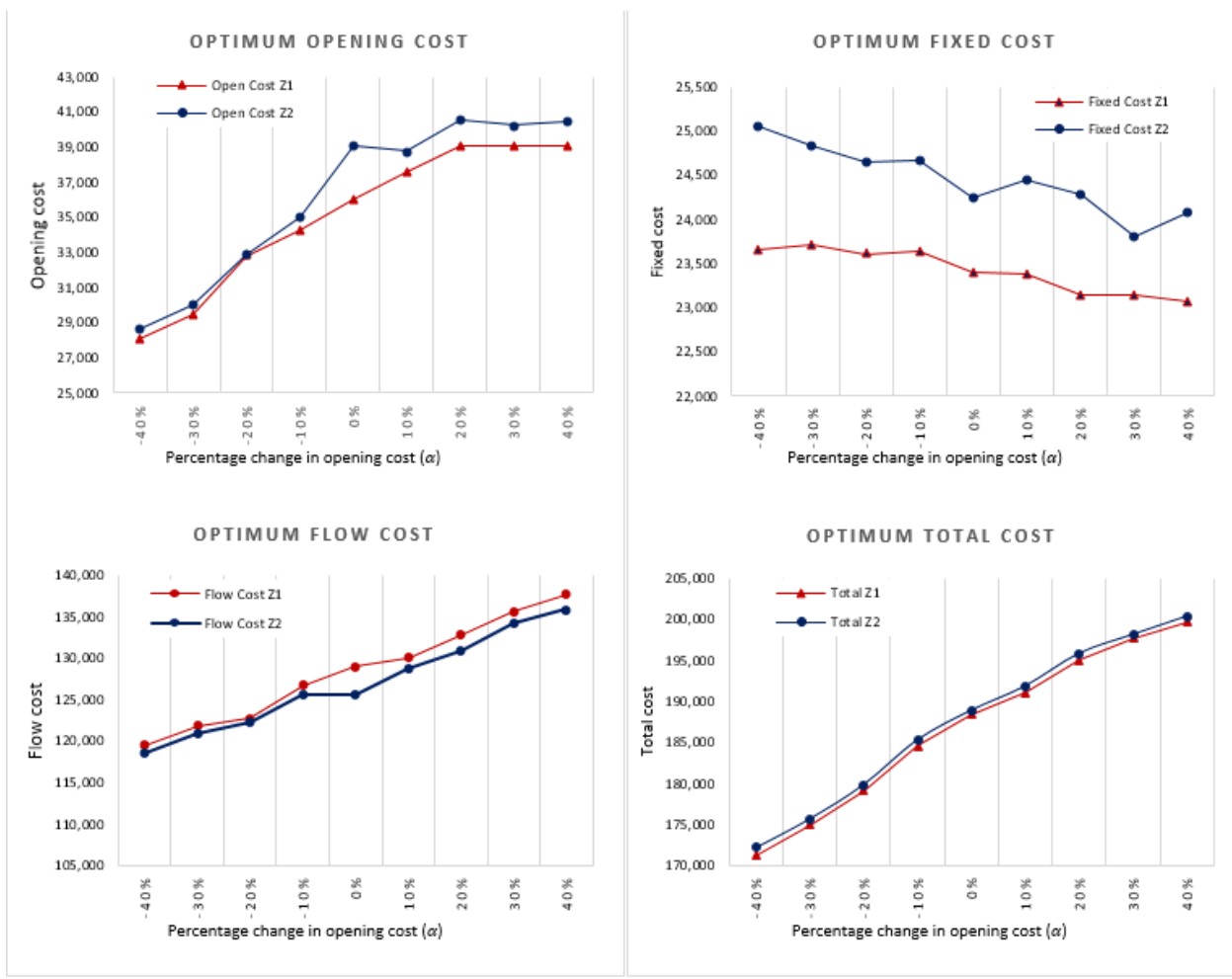

**Figure 6.** Impact of opening cost change on the output of the two models.

### 5.2. Variable Costs

In order to investigate the impact of variable costs on the two optimization models, another thirty problem instances of the same size were generated uniformly for each $\alpha$ by changing the variable cost from the ranges shown in Table 4. From the table, the variable cost was generated from the range [$10 \times (1 + \alpha)$ to $30 \times (1 + \alpha)$] for the first stage and [$10 \times (1 + \alpha)$ to $50 \times (1 + \alpha)$] for the second stage. Where $\alpha$ takes the same values (from −40% to 40%). The fixed cost generation was affected by changing the variable cost, where the fixed cost value was multiplied by the average of the variable cost, and the generated problems were solved optimally using the two models.

**Table 4.** Variable cost sensitivity analysis.

| Parameter | Range | |
|---|---|---|
| Retailer demand | 50 | 500 |
| Opening cost of the distribution center. | 10,000 | 20,000 |
| Fixed cost = 30–50 × average variable cost | 30 | 50 |
| Variable cost = 10–30 in stage 1 | $10(1+\alpha)$ | $30(1+\alpha)$ |
| Variable cost = 10–50 in stage 2 | $10(1+\alpha)$ | $50(1+\alpha)$ |

### 5.3. Discussion

Figure 6 shows that model $Z_2$, which disregarded FCs in minimization, performed better on flow costs alone. However, model $Z_1$ outperformed model $Z_2$ in the opening, fixed, and total costs. Figure 7 shows the average results of opening, fixed, flow, and total costs when changing variable costs. As we can see from the figure, $Z_2$ gives better performance on the opening and fixed costs, but it has no effect on flow and total cost output. The percentage improvement of $Z_1$ over $Z_2$ is shown in Figure 8. We can notice from the figure that as $\alpha$ increases, the relative percentage deviation of the two models increases with increasing the variable cost, which affects the FC, and decreases with increasing the opening cost. Consequently, it can be concluded that fixed costs significantly impact supply chain network design.

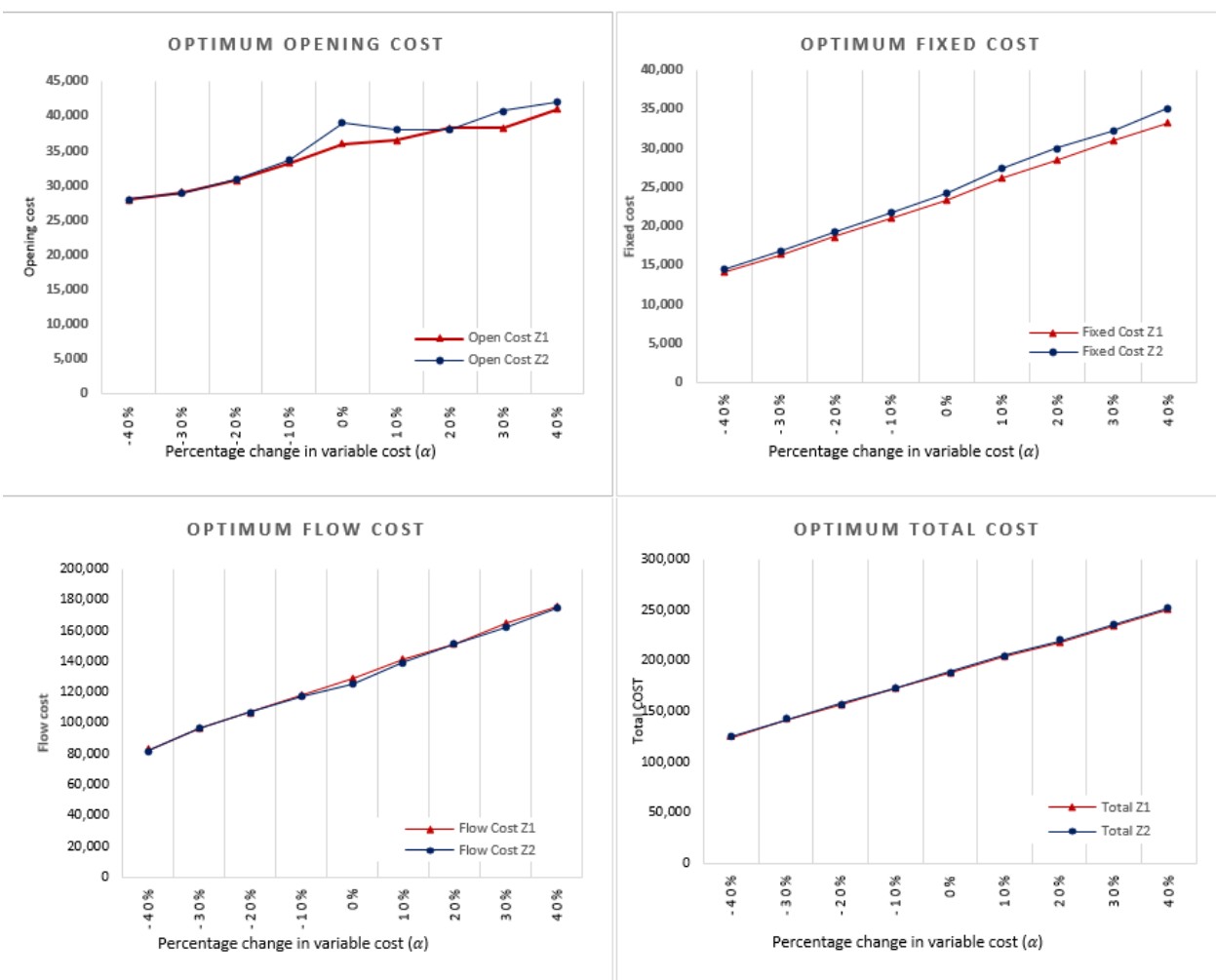

**Figure 7.** Impact of variable cost change on the output of the two models.

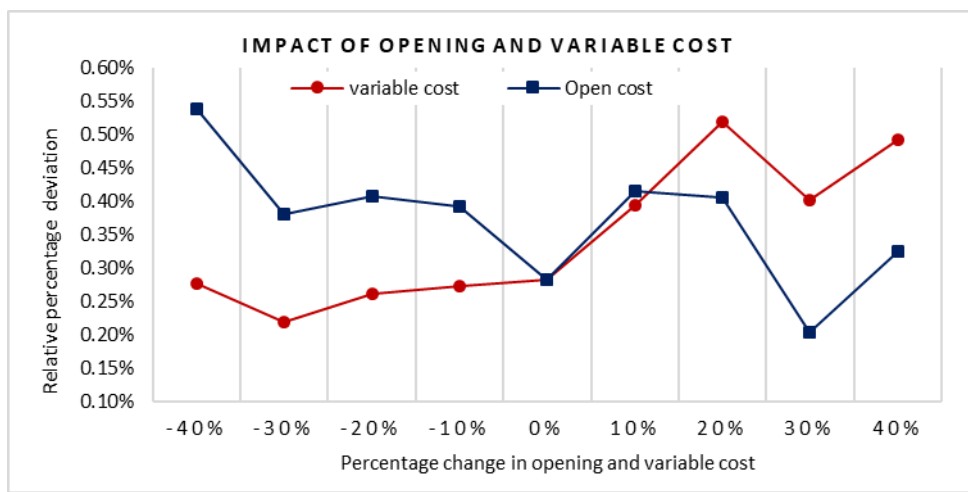

**Figure 8.** Impact of opening and variable costs on the output of the two models.

## 6. Conclusions and Future Work

This paper examined the impact of the fixed cost increase as well as the variable cost on the optimization of the two-stage supply chain network design. A numerical study on two models, $Z_1$ and $Z_2$, was designed and used to solve the considered group of instances. The first model, $Z_1$, considered both fixed and variable costs besides the opening cost in minimization, whereas the second one, $Z_2$, considered only variable costs and opening costs in minimization. To achieve the global optimum, the problem was formulated as a mixed-integer nonlinear programming model and coded using Lingo19. In order to demonstrate the impact of the fixed cost increase on the proposed models, four groups of network characteristics with different sizes were designed, and 260 instances were randomly generated for each network. Lingo 19 was used to solve the 1040 instances dataset for the first and second models. The results of the two models were compared using the average percentage relative deviation. In addition, a sensitivity analysis was conducted to determine the effect of varying opening and variable costs on the optimization. Computational results revealed that the optimal solutions of the first model $Z_1$ (when considering FCs in optimization) outperformed the other optimal solution of the second model $Z_2$ (when not considering FCs in optimization). Therefore, fixed costs are of paramount importance in the optimal design of supply chains.

For future studies, the single-product model in the current study has two stages, which can be developed into a multiproduct and/or multistage network. In contrast, the uncertainty included in capacity, demand, and cost could be considered for practical applications.

**Author Contributions:** Conceptualization, K.M. and R.E.; methodology, R.E.; software, A.M. and R.E.; validation, K.M. and R.E.; formal analysis, R.E.; investigation, R.E.; data curation, R.E.; writing—original draft preparation, A.M.; writing—review and editing, K.M. and R.E.; visualization, K.M. and R.E.; supervision, R.E.; funding acquisition, R.E. All authors have read and agreed to the published version of the manuscript.

**Funding:** This research was funded by the Deanship of Scientific Research, King Khalid University, Kingdom of Saudi Arabia, and the grant number is GRP/202/44.

**Institutional Review Board Statement:** Not applicable.

**Informed Consent Statement:** Not applicable.

**Data Availability Statement:** The original contributions presented in the study are included in the article. Further inquiries can be directed to the corresponding author.

**Acknowledgments:** We would like to express our gratitude to the Deanship of Scientific Research, King Khalid University, Kingdom of Saudi Arabia, for funding this work, as well as to family, friends, and colleagues for their constant inspiration and encouragement.

**Conflicts of Interest:** The authors declare no conflict of interest.

## Appendix A

The Lingo code of the first model, Z1. This code is applied to the generated data instances of group G_1 (2–5–10). The codes of the other groups and all results are provided in the article.

The Lingo model as shown below:

```
MODEL:
SETS:
PLANT:PLANTCAPACITY;
DISTRIBUTOR:DISTRIBUTORCAPACITY,OPINNINGCOST,U;
RETAILER:DEMAND;
LINK1(PLANT,DISTRIBUTOR):FIJ,YIJ,CIJ,XIJ;
LINK2(DISTRIBUTOR,RETAILER):FJK,YJK,CJK,XJK;
ENDSETS

DATA:
PLANT = P1,      P2;
DISTRIBUTOR = D1,       D2,    D3,     D4      D5;
RETAILER = R1,  R2,     R3,     R4,     R5,     R6      R7      R8      R9      R10;
PLANTCAPACITY = @file('PDR_2-5-10_1.txt');
DEMAND  = @file('PDR_2-5-10_1.TXT');
OPINNINGCOST = @file('PDR_2-5-10_1.TXT');
FIJ  = @file('PDR_2-5-10_1.TXT');
CIJ  = @file('PDR_2-5-10_1.TXT');
FJK  = @file('PDR_2-5-10_1.TXT');
CJK  = @file('PDR_2-5-10_1.TXT');

@TEXT('OUT1.TXT') = @WRITE('   PLANT       DISTRIBUTOR      QUANTITY', @NEWLINE( 1));
@TEXT('OUT1.TXT') = @WRITE('   ----------------------------', @NEWLINE( 1));
@TEXT('OUT1.TXT') = @WRITEFOR( LINK1( I, J) | XIJ( I, J) #GT# 0:8*' ', PLANT( I), 8*' ',DISTRIBUTOR( J), 8*' ', @FORMAT( XIJ( I, J), '8.0f'),
@NEWLINE( 1));
@TEXT('OUT1.TXT') = @WRITE('   ----------------------------', @NEWLINE( 2));
@TEXT('OUT1.TXT') = @WRITE('   DISTRIBUTOR    RETAILER      QUANTITY', @NEWLINE( 1));
@TEXT('OUT1.TXT') = @WRITE('   ----------------------------', @NEWLINE( 1));

@TEXT('OUT1.TXT') = @WRITEFOR( LINK2( J,K) | XJK( J,K) #GT# 0:8*' ', DISTRIBUTOR( J), 8*' ', RETAILER( K), 8*' ', @FORMAT( XJK( J,K),
'8.0f'), @NEWLINE( 1));
@TEXT('OUT1.TXT') = @WRITE('   ----------------------------', @NEWLINE( 1));

ENDDATA

[TCOST]  MIN  = FixedCost  + FlowCost + OpeningCost ;
        FixedCost = @SUM(LINK1(I,J):FIJ(I,J)*YIJ(I,J)) + @SUM(LINK2(J,K):FJK(J,K)*YJK(J,K)) ;
        FlowCost  = @SUM(LINK1(I,J):CIJ(I,J)*XIJ(I,J)) + @SUM(LINK2(J,K):CJK(J,K)*XJK(J,K)) ;
        OpeningCost = @SUM(DISTRIBUTOR(J):OPINNINGCOST(J)*U(J));

@FOR(PLANT(I): @SUM(DISTRIBUTOR(J):XIJ(I,J))< PLANTCAPACITY(I));
@FOR(DISTRIBUTOR(J):@SUM(RETAILER(K):XJK(J,K))<= DISTRIBUTORCAPACITY(J)*U(J));
@FOR(RETAILER(K):@SUM(DISTRIBUTOR(J):XJK(J,K))= DEMAND(K));
@FOR(DISTRIBUTOR(J):@SUM(RETAILER(K):DEMAND(K))= DISTRIBUTORCAPACITY(J));
! DC balance constraints;
@FOR(DISTRIBUTOR(J):@SUM(PLANT(I):XIJ(I,J)) = @SUM(RETAILER(K):XJK(J,K)));

@FOR(LINK1:XIJ> 0);
@FOR(LINK2:XJK> 0);
@FOR(LINK1:YIJ = @IF(XIJ#GT#0,1,0));
@FOR(LINK2:YJK = @IF(XJK#GT#0,1,0));
@FOR(DISTRIBUTOR:@BIN(U));

!TOTAL COST OUTPUT;
CALC:
  @SOLVE();
   @TEXT('CostandTime_2-5-10-Z1-Obj.TXT', 'a') = @WRITE( 'PDR_2-5-10_1.TXT',6*' ',   OpeningCost,6*' ', FixedCost ,6*' ', FlowCost ,6*'
,TCOST, 6*' ','CPU = ',6*' ', @Time(), @NEWLINE( 1));
ENDCALC

END

ALTER ALL '1.txt'3_%10-%-50.txt'
GO
ALTER ALL '3_%10-%-50.txt'3_%10-%-40.txt'
GO
ALTER ALL '3_%10-%-40.txt'3_%10-%-30.txt'
GO
ALTER ALL '3_%10-%-30.txt'3_%10-%-20.txt'
GO
...
```

**Appendix B**

The Lingo code of the second model, Z2. This code is applied to the generated data instances of group G_1 (2-5-10). The codes of the other groups and all results are provided in the article.

The Lingo model as shown below:

```
MODEL:
SETS:
PLANT:PLANTCAPACITY;
DISTRIBUTOR:DISTRIBUTORCAPACITY,OPINNINGCOST,U;
RETAILER:DEMAND;
LINK1(PLANT,DISTRIBUTOR):FIJ,YIJ,CIJ,XIJ;
LINK2(DISTRIBUTOR,RETAILER):FJK,YJK,CJK,XJK;
ENDSETS

DATA:
PLANT = P1,      P2;
DISTRIBUTOR = D1,      D2,      D3,      D4      D5;
RETAILER = R1,    R2,      R3,      R4,      R5,      R6      R7      R8      R9      R10;
PLANTCAPACITY = @file('PDR_2-5-10_1.txt');
DEMAND   = @file('PDR_2-5-10_1.TXT');
OPINNINGCOST = @file('PDR_2-5-10_1.TXT');
FIJ   = @file('PDR_2-5-10_1.TXT');
CIJ   = @file('PDR_2-5-10_1.TXT');
FJK = @file('PDR_2-5-10_1.TXT');
CJK   = @file('PDR_2-5-10_1.TXT');

@TEXT('OUT1.TXT') = @WRITE('    PLANT      DISTRIBUTOR      QUANTITY', @NEWLINE( 1));
@TEXT('OUT1.TXT') = @WRITE('      ---------------------', @NEWLINE( 1));
@TEXT('OUT1.TXT')  = @WRITEFOR( LINK1( I, J) | XIJ( I, J) #GT# 0:8*' ', PLANT( I), 8*' ',DISTRIBUTOR( J), 8*' ', @FORMAT( XIJ( I, J), '8.0f'),
@NEWLINE( 1));
@TEXT('OUT1.TXT') = @WRITE('      ---------------------', @NEWLINE( 2));
@TEXT('OUT1.TXT') = @WRITE('    DISTRIBUTOR    RETAILER      QUANTITY', @NEWLINE( 1));
@TEXT('OUT1.TXT')  = @WRITE('      ---------------------', @NEWLINE( 1));

@TEXT('OUT1.TXT')  = @WRITEFOR( LINK2( J,K) | XJK( J,K) #GT# 0:8*' ', DISTRIBUTOR( J), 8*' ', RETAILER( K), 8*' ', @FORMAT( XJK( J,K),
'8.0f'), @NEWLINE( 1));
@TEXT('OUT1.TXT')  = @WRITE('      ---------------------', @NEWLINE( 1));

ENDDATA

[TCOST]  MIN  = FlowCost + OpeningCost ;
         FixedCost = @SUM(LINK1(I,J):FIJ(I,J)*YIJ(I,J)) + @SUM(LINK2(J,K):FJK(J,K)*YJK(J,K)) ;
         FlowCost = @SUM(LINK1(I,J):CIJ(I,J)*XIJ(I,J)) + @SUM(LINK2(J,K):CJK(J,K)*XJK(J,K)) ;;
         OpeningCost = @SUM(DISTRIBUTOR(J):OPINNINGCOST(J)*U(J));

@FOR(PLANT(I): @SUM(DISTRIBUTOR(J):XIJ(I,J))< PLANTCAPACITY(I));
@FOR(DISTRIBUTOR(J):@SUM(RETAILER(K):XJK(J,K))<=  DISTRIBUTORCAPACITY(J)*U(J));
@FOR(RETAILER(K):@SUM(DISTRIBUTOR(J):XJK(J,K))=  DEMAND(K));
@FOR(DISTRIBUTOR(J):@SUM(RETAILER(K):DEMAND(K))= DISTRIBUTORCAPACITY(J));
! DC balance constraints;
@FOR(DISTRIBUTOR(J):@SUM(PLANT(I):XIJ(I,J)) = @SUM(RETAILER(K):XJK(J,K)));

@FOR(LINK1:XIJ> 0);
@FOR(LINK2:XJK> 0);
@FOR(LINK1:YIJ = @IF(XIJ#GT#0,1,0));
@FOR(LINK2:YJK = @IF(XJK#GT#0,1,0));
@FOR(DISTRIBUTOR:@BIN(U));

!TOTAL COST OUTPUT;
CALC:
   @SOLVE();
@TEXT('CostandTime_2-5-10-Z2-Obj.TXT','a') = @WRITE('PDR_2-5-10_1.TXT',6*'   OpeningCost,6*' ', FixedCost ,6*' ', FlowCost ,6*' ',Fixed-
Cost  + TCOST, 6*' ','CPU = ',6*' ',@Time(),  @NEWLINE( 1));

ENDCALC
END
ALTER ALL '1.txt'3_%10-%-50.txt'
GO
ALTER ALL '3_%10-%-50.txt'3_%10-%-40.txt'
GO
ALTER ALL '3_%10-%-40.txt'3_%10-%-30.txt'
GO
ALTER ALL '3_%10-%-30.txt'3_%10-%-20.txt'
GO
...
```

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
