# Peer review of "Impact of Fixed Cost Increase on the Optimization of Two-Stage Sustainable Supply Chain Networks"

_sustainability, doi:10.3390/su151813949_

Round 1
Reviewer 1 Report
This paper develops an optimization model on two-stage sustainable supply chain network design by addressing the influence of fixed cost. The impact of fixed costs on the optimal design of two-stage fixed cost transportation problem is investigated. The paper is well organized, and it can be published after following issues are resolved.
(1) The research motivations and research gaps should be addressed to help readers better understand the theoretical contributions.
(2)The literature review section needs to be re-arranged by categrizing references instead of listing all separated publications. Besides, a systematic review study should be performed.
(3)The algorithm section is missing, which algorithm is employed to deal with the formulated nonlinear programming model.
(4)It will be better if research implications could be addressed in the discussion section.
Author Response
Comment 1:
(1) The research motivations and research gaps should be addressed to help readers better understand the theoretical contributions.
Answer:
The abstract of the paper has been revised. Although most researchers have used fixed costs in supply chain design, none has studied their impact on optimal supply chain design. The design of the network can change when we take fixed costs into optimization. The literature demonstrates that the researchers consider FC when developing supply chains without study of its impact on the optimization, which is the motivation of this study.
Comment 2:
(2) The literature review section needs to be re-arranged by categorizing references instead of listing all separated publications. Besides, a systematic review study should be performed.
Answer:
The literature review was revised and re-arranged by categorizing references.
Comment 3:
(3) The algorithm section is missing, which algorithm is employed to deal with the formulated nonlinear programming model.
Answer:
The problem is formulated as a mixed-integer nonlinear programming model and coded using Lingo19 software to achieve the global optimum.
Comment 4:
(4) It will be better if research implications could be addressed in the discussion section.
Answer:
The optimal solutions are improved when considering FC in the optimization and the following is observed: (1) the average percentage of the total cost is enhanced by increasing the number of routes β whose FC increased. (2) For any specific change in the number of routes (x-axis) whose fixed costs will change, the percentage of improvement is directly proportional to the percentage change in FC. Therefore, computational results shown in Figures (2 – 5) reveal that the optimal solutions of the first model outperforms the other optimal solutions of the second model , and FC increase affect the optimality.
Reviewer 2 Report
The authors' work displays promise; however, several concerns necessitate resolution.
The abstract should be revised for clarity, effectively emphasizing the paper's significant contributions.
The study's progression of the existing field of knowledge and novel insights remains unclear. Clarification on the novelty could be achieved by explicitly addressing research gaps.
It is advisable to enhance the introduction by providing a more comprehensive research roadmap. This inclusion would aid readers in navigating the research process and paper modules.
Enhancements to the literature review are recommended, including the incorporation of various studies from 2019 to 2023.
The method section's description appears convoluted, necessitating clarification of fundamental concepts and definitions.
To enhance the conclusion, it is recommended to emphasize the core achievements of the research, principal managerial insights, and novel prospects for the future.
Author Response
Comment 1:
The abstract should be revised for clarity, effectively emphasizing the paper's significant contributions.
Answer:
The abstract of the paper has been revised. Although most researchers have used fixed costs in supply chain design, none has studied their impact on optimal supply chain design. The design of the network can change when we take fixed costs into optimization. The literature demonstrates that the researchers consider FC when developing supply chains without study of its impact on the optimization, which is the motivation of this study.
Comment 2:
The study's progression of the existing field of knowledge and novel insights remains unclear. Clarification on the novelty could be achieved by explicitly addressing research gaps.
Answer:
Although most researchers have used fixed costs in supply chain design, none has studied their impact on optimal supply chain design. The design of the network can change when we take fixed costs into optimization. The literature demonstrates that the researchers consider FC when developing supply chains without study of its impact on the optimization, which is the motivation of this study.
Comment 3:
It is advisable to enhance the introduction by providing a more comprehensive research roadmap. This inclusion would aid readers in navigating the research process and paper modules.
Answer:
The structure of the research is as follows: section 2 presents a literature review. In section 3, the problem formulation is presented. In section 4, the results and computational study are illustrated. Sensitivity analysis is described in section 5. Section 6 provides conclusions and suggestions for future work.
Comment 4:
Enhancements to the literature review are recommended, including the incorporation of various studies from 2019 to 2023.
Answer:
The literature review was revised and updated with up-to-date references (2019-2023).
Comment 5:
The method section's description appears convoluted, necessitating clarification of fundamental concepts and definitions.
Answer:
The fundamental concepts and definitions of the problem are illustrated in page 6-17 and the graphical representation of the proposed network shown in figure 1.
Comment 6:
To enhance the conclusion, it is recommended to emphasize the core achievements of the research, principal managerial insights, and novel prospects for the future.
Answer:
The conclusion was enhanced, managerial insights fixed costs are of paramount importance in the optimal design of supply chains.
Reviewer 3 Report
- In Section 1, unfortunately I can’t see research gap and shortcoming of previous research. While in this section the focus goes on research gap and its justifying. I would like to see some debate here.
- In section 1, after justifying the research gap, I would like to see some research questions. The authors explained the problem but I would like to see some debate here, comparing similar work and the novelty of this work.
- At page 4, Multi-Stage Multi-Period Fixed Charge Transportation Problem, add abbreviation in parenthesis (MPFCTP). As you did in continue.
- Better you draw a SCN picture for bolding your problem.
- Please justify how many decision variable you got.
- Before conclusion section it is better you add some discussion about your research questions and managerial implication of your work which too weak. Please strengthen this section. Add some research limitations in conclusion section if available.
- In Section 1, unfortunately I can’t see research gap and shortcoming of previous research. While in this section the focus goes on research gap and its justifying. I would like to see some debate here.
- In section 1, after justifying the research gap, I would like to see some research questions. The authors explained the problem but I would like to see some debate here, comparing similar work and the novelty of this work.
- At page 4, Multi-Stage Multi-Period Fixed Charge Transportation Problem, add abbreviation in parenthesis (MPFCTP). As you did in continue.
- Better you draw a SCN picture for bolding your problem.
- Please justify how many decision variable you got.
- Before conclusion section it is better you add some discussion about your research questions and managerial implication of your work which too weak. Please strengthen this section. Add some research limitations in conclusion section if available.
Minor editing of English language required
Author Response
the response report is attached

Round 2
Reviewer 2 Report
Authors have improved the paper well based on my comments and it can be accepted in its current form